# How Tokenization Limits Phonological Knowledge Representation in Language Models and How to Improve Them

**Disen Liao** [1]  **Freda Shi** [1]

## Abstract

Tokens serve as the fundamental units in language models (LMs) for processing input, generated through the process of tokenization. Tokenization can split a word into multiple subwords, a process that differs significantly from how humans perceive words, particularly in phonology. In this work, we examine two types of phonological features: local phonological coherence and prosodic structure. Using probing techniques, we demonstrate that tokenization impairs LMs' ability to capture phonological features. Furthermore, we show that tokenization affects LMs' inference results, which is one of their primary applications. Finally, we propose a data-efficient fine-tuning approach for large language models (LLMs) that leverages their pre-trained pronunciation knowledge, significantly enhancing inference performance on phonology-related tasks while preserving the model's ability on other tasks.

## 1. introduction

With the rapid advancement of language models (LMs), even those trained solely on text data, powerful models like GPT-4o appear to exhibit nontrivial knowledge about word pronunciations. This capability has led to their use in poetry generation (Zhang & Eger, 2024; Yu et al., 2024) and language learning (Hamaniuk, 2021; Bonner et al., 2023). However, how text-only LMs represent and process word sounds remains unclear.

In this work, we aim to provide insights into how phonological information is encoded in LMs by analyzing their performance on three related tasks, including: (1) **Rhyming Awareness**, which determines whether two words share the same ending sound; (2) **Grapheme-to-Phoneme (G2P)**

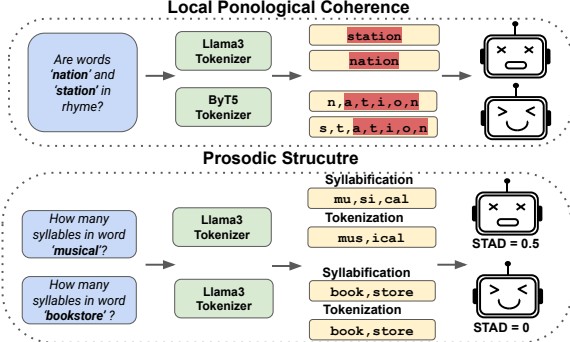

*Figure 1.* **Top:** Character-based tokenization (e.g., ByT5) provides finer-grained segmentation than subword tokenization (e.g., Llama3), aiding LMs in capturing local phonological coherence, such as rhyming patterns. **Bottom:** LMs exhibit better phonological understanding when tokenization aligns with syllabification (STAD = 0), whereas misalignment (STAD = 0.5) impairs performance in prosodic structure tasks.

conversion, which transcribes a word into its ARPAbet representation, a phonetic transcription system widely used in text-to-speech tasks; and (3) **Syllable Counting**, which identifies the number of syllables in a word. Through experiments involving probing hidden states and direct inference on these tasks, we find that tokenization, the first processing step in every LM, plays a crucial role in how LMs understand word sounds.

Common tokenization algorithms include Byte Pair Encoding (BPE; Gage, 1994; Sennrich et al., 2015) and UnigramLM (Kudo, 2018), where they segment the input into subwords and assign IDs to these subwords, maintaining a manageable vocabulary size. However, there are clear limitations of this approach. First, prior work has challenged subword tokenization in lexical and arithmetic tasks (Singh & Strouse, 2024; Bunzeck et al., 2024), and alternative models like ByT5 (Xue et al., 2022), which use character-level tokenization, have demonstrated greater robustness to noise and superior performance in spelling and pronunciation tasks. Second, generalizing to out-of-vocabulary words remains challenging due to misalignment with morpheme boundaries (Batsuren et al., 2024). In line with existing work, our findings suggest that tokenization affects phono-

[1]David R. Cheriton School of Computer Science, University of Waterloo, Waterloo, ON, Canada. Correspondence to: Disen Liao <d7liao@uwaterloo.ca>.

*Non-archival presentation at ICML 2025 Tokenization Workshop (TokShop)*, Vancouver, Canada. 2025.

logical understanding in two key ways: (1) finer-grained tokenization methods, such as character-level tokenization, improve the model's ability to capture **local phonological coherence** (e.g., rhyming awareness); and (2) tokenization schemes that align with a word's syllabification enhance the model's ability to learn **prosodic structure**, benefitting tasks like G2P and syllable counting. To quantify the alignment between syllabification and tokenization, we introduce a metric called the **Syllabification-Tokenization Alignment Distance (STAD)**. A lower STAD score indicates better alignment, with a score of 0 denoting perfect correspondence between syllable boundaries and token boundaries. Our results suggest that tokenization introduces systematic biases: language models exhibit improved performance on rhyming awareness tasks when words are tokenized at a finer granularity, while tasks dependent on prosodic structures benefit from words with low STAD scores.

To mitigate the phonological biases introduced by tokenization, we propose an efficient data creation method for instruct-tuning large language models (LLMs) (>7B parameters). Leveraging the model's existing knowledge of the International Phonetic Alphabet (IPA), we fine-tune it to utilize IPA for phonology-related tasks, leading to performance improvements across all three evaluated tasks. Finally, we analyze words with high and low STAD scores, providing linguistic explanations for the observed differences. In summary, our contributions are:

- We identify potential issues in LMs' tokenization that hinder phonological understanding, as revealed through probing hidden layers. We propose the **Syllabification-Tokenization Alignment Distance (STAD)** metric to quantify deviations between tokenization and syllabification.

- We introduce a data augmentation strategy to fine-tune LLMs for phonology-related tasks using IPA. This approach significantly enhances task performance with minimal data while preserving the model's general performance on other tasks.

## 2. Related Work

**Probing.** Probing (Ettinger et al., 2016) investigates the internal representations of language models by training lightweight classifiers on hidden states to predict specific attributes, such as truthfulness (Azaria & Mitchell, 2023), spatial understanding (Gurnee & Tegmark, 2023), sound perception (Ngo & Kim, 2024), and sound symbolism (Alper & Averbuch-Elor, 2024). Compared to performance-based evaluation methods like accuracy, probing reveals more nuanced latent knowledge (i.e., competence; Chomsky, 1965) embedded within a model's internal activations (Burns et al., 2022)—even when a model produces incorrect predictions, it may still encode relevant information. In particular, Burns

et al. (2022) introduced contrast-consistent search (CSS), a method that maps the hidden states of true and false statements to probabilities and trains these probabilities to achieve "contrast consistency." In our work, probing allows us to evaluate smaller LMs that lack question-answering capabilities by measuring the performance of trained classifiers. Kaushal & Mahowald (2022) employed probing methods to demonstrate LMs. In contrast, our work uses probing to investigate how subword tokenization may obscure the encoding of local phonological features. Prior studies have advocated for the use of simple linear models in probing tasks (Alain & Bengio, 2018; Ettinger et al., 2016; Hewitt & Manning, 2019), arguing that less expressive classifiers provide more interpretable insights into the representations learned by the model. Motivated by this, we adopt simple linear models—specifically, logistic regression and ridge regression—in our experiments to maintain a clear separation between the model's representational capacity and the complexity of the probing classifier.

**LM Phonology.** Benchmarks for assessing the phonological capabilities of LMs are still in an early stage. Recently, Suvarna et al. (2024) introduced a benchmark specifically designed to evaluate LLMs' performance on phonological tasks. They proposed three tasks: Rhyming Generation, G2P, and Syllable Counting to evaluate the phonology ability of LLMs, our chosen tasks are inspired by their design. Concurrently, using LMs for phonology task is a promising direction, some phoneme-based models have been tailored for lower-level phonological tasks. For instance, PhonemeBERT (Sundararaman et al., 2021) was fine-tuned on a dataset combining ASR transcripts and phonemes, while Mix-Phoneme BERT (Zhang et al., 2022) was pre-trained with phonemes and sub-phonemes as additional features to enhance text-to-speech performance. Furthermore, (Qharabagh et al., 2024) demonstrated that LLMs could significantly improve grapheme-to-phoneme conversion, especially in low-resource languages, underscoring the potential of LLMs to advance phonological processing in linguistically underserved contexts.

**Tokenization Pitfalls.** Subword-based tokenization algorithms, such as BPE, are widely used in training contemporary LLMs. Prior research has highlighted how tokenization can introduce artifacts that impact model performance, particularly in tasks involving phoneme and grapheme representations. Shin et al. (2020) found that certain tokens can negatively affect LMs' performance. Additionally, tokenization consistency plays a crucial role in extractive QA tasks (Sun et al., 2023). Singh & Strouse (2024) further argued that for numeric reasoning tasks, LLMs perform better when numbers are tokenized from right to left. To mitigate tokenization-induced issues, Deng et al. (2023) proposed a *rephrase-and-respond* approach, which aligns with our IPA fine-tuning strategy—incorporating additional information

to circumvent tokenization pitfalls. Meanwhile, character-level tokenization has been explored as an alternative to subword-based methods to eliminate tokenization biases. For instance, BERT has been shown to exhibit sensitivity to misspellings due to its reliance on subword tokenization (Sun et al., 2020). Bunzeck et al. (2024) retrained a smaller language model using grapheme- and phoneme-based tokenization, demonstrating that these approaches can achieve comparable performance on tasks such as lexical decision and rhyme prediction. To address the limitations of subword tokenization, character-level tokenization strategies have been explored, including pre-training variants such as CANINE (Clark et al., 2022) and ByT5 (Xue et al., 2022). Our work highlights tokenizer pitfalls in phonological tasks, extending these findings to phoneme and grapheme representations.

## 3. How Tokenization Affects Phonological Competence

To investigate how LMs represent the sound of words, we employ probing to analyze their hidden state representations. Formally, given an LM $f$, we prompt it with text $P$, which can be tokenized into $k$ subwords $\{w_1, \ldots, w_k\}$, and model $f$ produces hidden states $\boldsymbol{h}_{il} \in \mathbb{R}^d$ for each subword token $i \in \{1, \ldots, k\}$ at each layer $\ell$, resulting in a hidden state matrix $\boldsymbol{H}_\ell = [\boldsymbol{h}_{1\ell}, \ldots, \boldsymbol{h}_{k\ell}]^T \in \mathbb{R}^{w \times d}$, where $d$ is the hidden dimension. To derive a fixed-size representation for the prompt $P$, it is common to either use the final layer hidden states or compute the average of the hidden states across layers or tokens. In our experiment, we will use the hidden state of the final token from the final layer, denoted $\boldsymbol{h}_\ell = \boldsymbol{h}_{k\ell}$ as the representation of the entire prompt $P$. Given a dataset of $n$ prompts with corresponding label $\boldsymbol{y} \in \mathbb{R}^n$, we extract $n$ such representations to construct our probing dataset:

$$\mathcal{D} = [\boldsymbol{h}_{1\ell}, \ldots, \boldsymbol{h}_{n\ell}]^T \in \mathbb{R}^{n \times d}$$

We then train a classifier or regressor (i.e., a probe) on $\mathcal{D}$ to predict the ground-truth labels $\boldsymbol{y}$ for downstream tasks, enabling us to analyze how well the LM encodes phonological information.

In this section, we analyze LM performance on phonological tasks that consist of one binary classification task—rhyming awareness—and two regression tasks—G2P and syllable counting—to examine the effect of tokenization on phonological processing. Our experiments reveal that tokenization influences how LMs encode phonology in two key ways: for phonological features that rely on **local phonological coherence**, finer-grained tokenization enhances model performance (Section 3.1); for features dependent on **prosodic structure**, alignment between tokenization and syllabification is particularly a crucial factor (Section 3.2).

### 3.1. Local Phonological Coherence: Fine-Grained Tokenization for Rhymes

| Model Size | Format | 0% | 20% | 40% | 60% | 80% | 100% |
|---|---|---|---|---|---|---|---|
| | | **Depth** (Accuracy ↑) | | | | | |
| Subword Tokenization | | | | | | | |
| **BERT** | | | | | | | |
| 110M | Orig | 56.0 | 67.6 | 68.3 | 70.9 | 71.0 | 70.5 |
| | Slash | **68.6** *** | **74.5** ** | **73.4** ** | **77.5** ** | **79.5** ** | **78.1** ** |
| **GPT2** | | | | | | | |
| 1.2B | Orig | 63.4 | 64.7 | 66.1 | 66.2 | 66.0 | 61.6 |
| | Slash | **71.7** *** | **76.9** *** | **77.2** *** | **79.1** *** | **78.5** *** | **77.5** *** |
| **GPT-neo-2.7b** | | | | | | | |
| 2.7B | Orig | 68.2 | 68.6 | 72.4 | 69.6 | 69.7 | 67.0 |
| | Slash | 73.2 | **82.5** *** | **83.9** *** | **82.5** *** | **81.6** *** | **82.4** *** |
| **Llama3-8b-Instruct** | | | | | | | |
| 8B | Orig | 71.4 | 80.7 | 76.6 | 76.8 | 70.5 | 78.4 |
| | Slash | 56.3 | **85.4** * | **81.9** ** | **77.9** | **71.9** | 76.1 |
| **Llama3.1-8b-Instruct** | | | | | | | |
| 8B | Orig | 72.5 | 79.8 | 79.0 | 77.9 | 77.3 | 74.9 |
| | Slash | 56.3 | **85.1** ** | **84.0** ** | **80.0** | **78.9** | **79.5** ** |
| **Mistral-7b-Instruct-v3** | | | | | | | |
| 7B | Orig | 64.5 | 80.6 | 80.8 | 78.8 | 77.4 | 74.7 |
| | Slash | 55.8 | **81.1** | **82.7** * | **79.5** | **79.0** * | **77.6** ** |
| Character Tokenization | | | | | | | |
| **ByT5-base** | | | | | | | |
| 580M | Orig | 45.5 | 79.6 | 81.0 | 79.9 | 80.3 | 66.3 |
| | Slash | - | - | - | - | - | - |
| **ByT5-small** | | | | | | | |
| 300M | Orig | 45.5 | 75.5 | 80.1 | 77.6 | 72.7 | 71.8 |
| | Slash | - | - | - | - | - | - |
| Control Experiment | | | | | | | |
| **Random Embeddings** | | | | | | | |
| - | - | 48.8 | 48.7 | 51.7 | 49.3 | 50.2 | 50.8 |

*Table 1.* Accuracy of logistic regression trained on language models of varying depths, comparing performance with words containing slash separators (Slash) vs. original words (Orig). The reported values are the average accuracy over 10 runs. **Bold** indicates that Slash outperforms Orig, while underlined values denote the highest accuracy in each row. A t-test is conducted to assess the hypothesis that Slash achieves higher accuracy than Orig, with significance levels indicated as follows: $p < 0.05$ (*), $p < 0.01$ (**), and $p < 0.001$ (***). We also include the results of 32-layers of randomized embeddings, and the results is almost random guess, meaning that our linear prober is not overfitting to the task.

To evaluate **local phonological coherence**, we use the **rhyming awareness** task, a fundamental phonological task that serves as an early indicator of phonological development in children with normal hearing (Bradley & Bryant, 1983) and a predictor of more complex phonological abilities (Adams, 1994). Rhyming awareness requires deter-

mining whether a given word pair $(w_1, w_2)$ rhymes, using a binary ground truth label $y$.

### 3.1.1. EXPERIMENT SETUP

**Dataset.** Since rhyming is often associated with grapheme similarity—where rhyming words typically share the same suffix—we aim to prevent LMs from taking shortcuts by relying solely on word endings. To achieve this, we construct a dataset consisting of 200-word pairs that rhyme but have different three-letter ending suffixes as positive pairs, along with 200 non-rhyming word pairs as negative pairs.

**Model.** We evaluate LMs with different architectures, sizes, and tokenization strategies: BERT (Koroteev, 2021), GPT-2 (Saphra & Lopez, 2019), GPT-neo-2.7B (Black et al., 2021), Llama3-8B, Llama3.1-8B (Grattafiori et al., 2024), Mistral-7B-v3 (Jiang et al., 2023). To compare against the character tokenization strategy, we also include ByT5-base and ByT5-small (Xue et al., 2022).

**Evaluation.** Since most tokenizers either retain a word as a single token or split it into multi-character subwords, they may fail to capture subtle intra-word features such as rhyme. We hypothesize that a more fine-grained tokenization approach improves an LM's capability to encode local phonological coherence. To test this, we modify the input words by inserting slashes between each consecutive pair of characters to enforce finer-grained tokenization. For example, while most tokenizers represent the word $w$ = "boy" as a single token, the transformed word $w'$ = "b/o/y" is tokenized as [`'b'`, `'/'`, `'o'`, `'/'`, `'y'`] for most tokenizers. This finer-grained segmentation lets the language models attend to pronunciation cues at a sub-word level. To ensure the effect is not tied to the slash delimiter alone, we reran the experiment with alternative punctuation marks (comma and period); see Appendix F for the results.

We split the dataset into 80% training and 20% test sets. To obtain hidden states, we construct two prompts: the original tokenization $P$ = "$w_1, w_2$" and finer-grained tokenization $P'$ = "$w'_1, w'_2$", then extract the corresponding hidden states $\boldsymbol{h}_\ell = f(P)$ and $\boldsymbol{h}'_\ell = f(P')$ from each layer $\ell$ of the LM. For each layer $\ell$, we train two logistic regression classifiers on $\boldsymbol{h}_\ell$ and $\boldsymbol{h}'_\ell$, respectively, and compare their performance on the test set. We present our results in Table 1.

### 3.2. Prosodic Structure: Alignment of Tokens and Syllables

Syllabification, also known as hyphenation, refers to dividing a word into its constituent syllables. A syllable is a unit of pronunciation that typically consists of a vowel sound, often accompanied by consonants, following linguistic rules that determine where natural breaks occur in spoken language. For instance, the word *decide* is syllabified as

[`'de'`, `'cide'`]. Understanding syllabification is essential for accurate pronunciation, linguistic analysis, and poetry. The rules governing syllable division vary across languages and depend on phonetic and morphological structures (Selkirk, 1986). The tokenization strategies of LMs, however, do not explicitly consider phonological principles. This discrepancy can lead to misalignment between tokenization and syllable boundaries, potentially hindering LMs' ability to capture **prosodic structure**—the broader phonological properties of words, including rhythm and syllable organization.

### 3.2.1. THE STAD SCORE

To quantify the deviation between tokenization and syllabification, we introduce a metric, syllabification-tokenization alignment distance (STAD). For a given word consisting of $n + 1$ characters $w = a_1 a_2 \ldots a_{n+1}$, there are $n$ possible positions to make a split. We use two binary vectors $\boldsymbol{v}_{\text{tok}} = [b_1, b_2, \ldots, b_n]$ and $\boldsymbol{v}_{\text{syl}} = [c_1, c_2, \ldots, c_n]$ to encode the splits. Here, $b_i, c_i \in \{0, 1\}$, where $b_i = 1$ indicates a tokenization split after the $i$-th character, and $c_i = 1$ indicates a syllabification split after the $i$-th character. For example, consider the word *musical*. According to syllabification rules, it splits into [`'mu'`, `'si'`, `'cal'`], represented as $\boldsymbol{v}_{\text{syl}} = [0, 1, 0, 1, 0, 0]$. Meanwhile, the tokenizer splits it as [`'mus'`, `'ical'`], yielding $\boldsymbol{v}_{\text{tok}} = [0, 0, 1, 0, 0, 0]$. The deviation between tokenization and syllabification is measured using the normalized Hamming distance (HD) between $\boldsymbol{v}_{\text{tok}}$ and $\boldsymbol{v}_{\text{syl}}$:

$$\text{STAD}(w) = \text{HD}(\boldsymbol{v}_{\text{tok}}, \boldsymbol{v}_{\text{syl}}) = \frac{\sum_{i=1}^{n} |b_i - c_i|}{n}.$$

Therefore, in the above example, the STAD score for *musical* is 0.5.

### 3.2.2. EXPERIMENT SETUP

**Dataset.** For each LM, we create two splits of words, the token-syllable aligned (A) split and the token-syllable misaligned (M) one. The words are sampled from google-10000-English,[1] which includes the 10,000 most frequent English words. For each LM, we sample 1,000 words with high STAD ($> 0.25$) as misaligned words and 1,000 aligned ones with 0 STAD. To probe the phonological understanding of LMs on different words, we consider two tasks: G2P and syllable counting.

**Models.** Similarly to Section 3.1, we experiment with various models. For this experiment, we add in BLOOM-560M (BigScience Workshop, 2022), Yi-6B (Young et al., 2024), Falcon-7B (Almazrouei et al., 2023), but exclude the two byte-level tokenization models.

---

[1] https://github.com/first20hours/google-10000-english

| Model | STAD | Syllable Counting ($R^2 \uparrow$) | | | | | | Grapheme-to-Phoneme ($R^2 \uparrow$) | | | | | |
|---|---|---|---|---|---|---|---|---|---|---|---|---|---|
| Align | | 0% | 20% | 40% | 60% | 80% | 100% | 0% | 20% | 40% | 60% | 80% | 100% |
| **BERT** | | | | | | | | | | | | | |
| A | 0.000 | ***
**0.999** | ***
**0.952** | 0.009 | 0.022 | 0.129 | 0.054 | 0.004 | 0.056 | 0.001 | 0.002 | **0.030** | 0.009 |
| M | 0.290 | 0.404 | 0.626 | 0.068 | 0.074 | 0.264 | 0.143 | 0.009 | 0.085 | 0.001 | 0.002 | 0.024 | 0.010 |
| **GPT2** | | | | | | | | | | | | | |
| A | 0.000 | 0.027 | ***
**0.980** | ***
**0.980** | ***
**0.969** | ***
**0.952** | ***
**0.929** | ***
**0.198** | ***
**0.229** | ***
0.232 | ***
**0.217** | ***
**0.185** | ***
**0.194** |
| M | 0.388 | 0.589 | 0.740 | 0.732 | 0.728 | 0.714 | 0.684 | 0.081 | 0.148 | 0.146 | 0.124 | 0.080 | 0.119 |
| **bloom-560m** | | | | | | | | | | | | | |
| A | 0.000 | 0.476 | ***
**0.947** | ***
0.966 | ***
**0.950** | ***
**0.936** | ***
**0.922** | 0.058 | **
**0.238** | *
**0.231** | **0.222** | **0.214** | **
**0.193** |
| M | 0.376 | 0.489 | 0.766 | 0.753 | 0.711 | 0.674 | 0.608 | 0.096 | 0.196 | 0.215 | 0.215 | 0.199 | 0.168 |
| **GPT-neo-2.7b** | | | | | | | | | | | | | |
| A | 0.000 | ***
**0.945** | ***
0.953 | ***
0.967 | ***
**0.942** | ***
**0.930** | ***
**0.914** | ***
**0.179** | **
0.219 | ***
**0.211** | ***
**0.148** | ***
**0.078** | ***
**0.004** |
| M | 0.388 | 0.555 | 0.787 | 0.758 | 0.692 | 0.634 | 0.539 | 0.072 | 0.169 | 0.111 | 0.005 | -0.124 | -0.219 |
| **gemma-7b** | | | | | | | | | | | | | |
| A | 0.000 | 0.383 | ***
**0.921** | ***
**0.934** | ***
0.976 | ***
**0.946** | *
**0.782** | ***
**0.168** | *
**0.279** | **0.247** | **0.260** | *
0.312 | **0.193** |
| M | 0.303 | 0.640 | 0.773 | 0.769 | 0.772 | 0.758 | 0.672 | 0.054 | 0.229 | 0.231 | 0.226 | 0.284 | 0.183 |
| **Llama3.1-8b-Instruct** | | | | | | | | | | | | | |
| A | 0.000 | 0.188 | ***
**0.936** | ***
0.939 | ***
**0.921** | ***
**0.898** | ***
**0.859** | 0.033 | *
**0.325** | **
**0.321** | **
0.387 | **
**0.357** | **0.166** |
| M | 0.372 | 0.211 | 0.783 | 0.789 | 0.769 | 0.754 | 0.723 | 0.029 | 0.304 | 0.285 | 0.349 | 0.317 | 0.157 |
| **Llama3-8b-Instruct** | | | | | | | | | | | | | |
| A | 0.000 | 0.152 | ***
**0.931** | ***
0.935 | ***
**0.923** | ***
**0.899** | ***
**0.860** | ***
**0.034** | ***
**0.349** | ***
**0.356** | ***
0.370 | ***
**0.366** | ***
**0.325** |
| M | 0.372 | 0.165 | 0.769 | 0.795 | 0.769 | 0.749 | 0.717 | 0.023 | 0.295 | 0.297 | 0.333 | 0.308 | 0.276 |
| **Mistral-7b-Instruct-v3** | | | | | | | | | | | | | |
| A | 0.000 | 0.028 | ***
**0.800** | ***
0.913 | ***
**0.911** | ***
**0.854** | ***
**0.816** | **0.001** | 0.212 | ***
0.301 | 0.314 | **0.297** | 0.239 |
| M | 0.348 | 0.045 | 0.708 | 0.804 | 0.806 | 0.789 | 0.762 | 0.000 | 0.214 | 0.283 | 0.317 | 0.282 | 0.261 |
| **Falcon3-7b-Instruct** | | | | | | | | | | | | | |
| A | 0.000 | 0.419 | ***
**0.974** | ***
0.977 | ***
**0.975** | ***
**0.975** | ***
**0.977** | ***
**0.100** | **
**0.209** | **
0.192 | ***
**0.153** | ***
**0.151** | ***
**0.149** |
| M | 0.337 | 0.618 | 0.776 | 0.769 | 0.734 | 0.728 | 0.729 | 0.050 | 0.148 | 0.155 | 0.054 | 0.004 | 0.025 |
| **Yi-1.5-6B-Chat** | | | | | | | | | | | | | |
| A | 0.000 | 0.252 | **
**0.925** | ***
**0.937** | ***
**0.940** | ***
0.941 | ***
**0.936** | 0.056 | **0.240** | ***
0.269 | *
**0.245** | **
**0.210** | **
**0.189** |
| M | 0.326 | 0.624 | 0.852 | 0.825 | 0.783 | 0.746 | 0.737 | 0.082 | 0.231 | 0.228 | 0.190 | 0.148 | 0.117 |
| Control Experiment | | | | | | | | | | | | | |
| **Randomized Embedding** | | | | | | | | | | | | | |
| - | - | -0.082 | -0.073 | 0.001 | -0.115 | -0.097 | -0.022 | -0.07 | -0.101 | -0.043 | -0.073 | -0.066 | -0.082 |

*Table 2.* $R^2$ of the ridge regression probe trained on hidden layers of varying depths for the G2P and syllable counting tasks, comparing performance between words with aligned syllables and tokens (A) and misaligned syllables and tokens (M). The reported values represent the average R-squared over 10 runs. **Bold** indicates that A outperforms M, while underlined values denote the highest R-squared in each row. A t-test is conducted to evaluate the hypothesis that words in group A achieve higher R-squared than those in group M, with significance levels indicated as follows: $p < 0.05$ (*), $p < 0.01$ (**), and $p < 0.001$ (***). We also include the control experiment where we randomly generate 32 layers of the embeddings.

**Evaluation.** For G2P task, we use the CMU Pronunciation Dictionary [2] as our reference standard. The library provides phoneme transcript for English words and the phonemes are given in ARPAbet, which consists of 39 different phonemes

[2] http://www.speech.cs.cmu.edu/cgi-bin/cmudict

to describe the pronunciation of a word. Compared to the International Phonetic Alphabet (IPA), ARPAbet offers a more practical representation for computational modeling: IPA contains a large set of symbols, many of which are difficult to encode consistently across systems, while ARPAbet uses a limited set of ASCII characters and is widely adopted in speech processing research. We encode the ARPAbet

transcript using indices 0 to 39, where index 0 is reserved for padding the encoded vector to make all encoded vector has the same length of 8, the maximum number of syllables in our dataset. For a word $w$, we form the prompt $P =$ '$w$', and extract the hidden states $\boldsymbol{h}_\ell = f(P)$ for each layer $\ell$. We represent the ARPAbet phoneme encoding of each word as a vector $\boldsymbol{y}_w \in \mathbb{R}^8$, where each entry corresponds to the index of a phoneme in the padded sequence. To map hidden state representations to phoneme sequences, we train a multi-label ridge regression model using the hidden states as input features and the phoneme indices as targets. We choose ridge regression over multi-class classification for this task because the latter requires training a separate classifier for each position with a 40-class output space, which is both computationally intensive and prone to overfitting given the limited size of training data. In contrast, ridge regression offers a simpler and more stable alternative that performs well in high-dimensional settings and provides smooth predictions suitable for downstream phoneme decoding.

For syllable counting, we use the same hidden states as the G2P task, and represent a label with an integer $y_w \in \mathbb{Z}_+$ indicating the number of syllables in the word $w$. Similarly, we fit the ridge regression on hidden states and the labels. We present the results for both experiments in Table 2.

### 3.3. Observations

In the rhyming awareness task (Table 1), we observe that a finer-grained tokenization strategy, achieved by inserting slashes within words, significantly enhances the local phonological coherence captured by LMs. Probes trained on hidden states from slash-inserted words exhibit substantially stronger predictive power across all layers beyond the word embedding layer for all tested LMs. Additionally, models employing character-level tokenization produce considerably more informative hidden states compared to similarly sized subword-tokenized models and achieve performance comparable to much larger subword-based models.

Furthermore, phonological features are more closely tied to morphosyntactic structures than to word semantics. Our findings resonate with that by Saphra & Lopez (2019), who suggest that early LM layers primarily encode syntactic features, while deeper layers capture semantic properties. Notably, in all three tasks, we found that phonology-related features were most expressively encoded in mid-range layers, typically spanning 20%–60% of the overall depth. Beyond this range, performance declined as deeper layers became increasingly associated with semantic processing.

To guard against the pitfall identified by Hewitt & Liang (2019)—namely, that an expressive probe can memorize the target function even when the representation lacks the relevant information—we replicate their "control" protocol for all three of our tasks (rhyming awareness, grapheme-to-phoneme, syllable counting): keeping the model's hidden states unchanged, we shuffle the labels to preserve marginal statistics and train the *same* linear probe on this nonsense task. As detailed in Appendix A, the probe's accuracy falls to chance on the binary task and its $R^2$ becomes zero or negative on the regression tasks across every layer and model, confirming that our linear prober is not powerful enough to invent the mapping on random data and that the positive results reported in the main paper genuinely reflect information encoded in the representations rather than overfitting by the probe itself.

## 4. How Tokenization Affects Inference

The probing experiments (Section 3) only demonstrate how LMs encode the phonology features of words, and our experiments suggest that tokenization plays an important role in LMs encoding those features. Currently, LMs are more used in scenarios where users directly get answers from the output of the LMs. In this section, we present experiments that suggest tokenization may affect the results of inference, but keeping the input in an appropriate format is more important in terms of the inference results. Also, we find that most LMs with large parameter sizes ($> 7B$) have a solid understanding of the IPA of the word, but the models will fail to adapt that knowledge in phonology-related tasks; therefore, we propose a data-efficiency way to fine-tune language models to improve the capability in phonology-related tasks.

### 4.1. Inference Using IPA

The International Phonetic Alphabet (IPA) is a more widely used and standardized representation of word pronunciation compared to the ARPAbet. LMs tend to exhibit a significantly better understanding of IPA than ARPAbet, making IPA a valuable tool for phonology-related tasks such as rhyme detection and G2P conversion. For example, consider the words *tough* and *though*. They share the same orthographic ending, "-ough," which might mislead an LM into classifying them as rhyming words. However, from their IPA transcriptions, /tʌf/ and /ðoʊ/, LMs can easily reveal that they do not rhyme. Despite possessing knowledge of IPA, LMs often fail to leverage it effectively in phonology-related tasks (Suvarna et al., 2024).

To address this limitation, we propose a data augmentation method to fine-tune LMs, ensuring they better utilize IPA representations in phonological tasks. Additionally, we carefully construct the QA training dataset to prevent the issue of *catastrophic forgetting* (Kirkpatrick et al., 2017), which can arise when models are fine-tuned on a specific domain, such as phonology, without maintaining generalization across broader linguistic tasks.

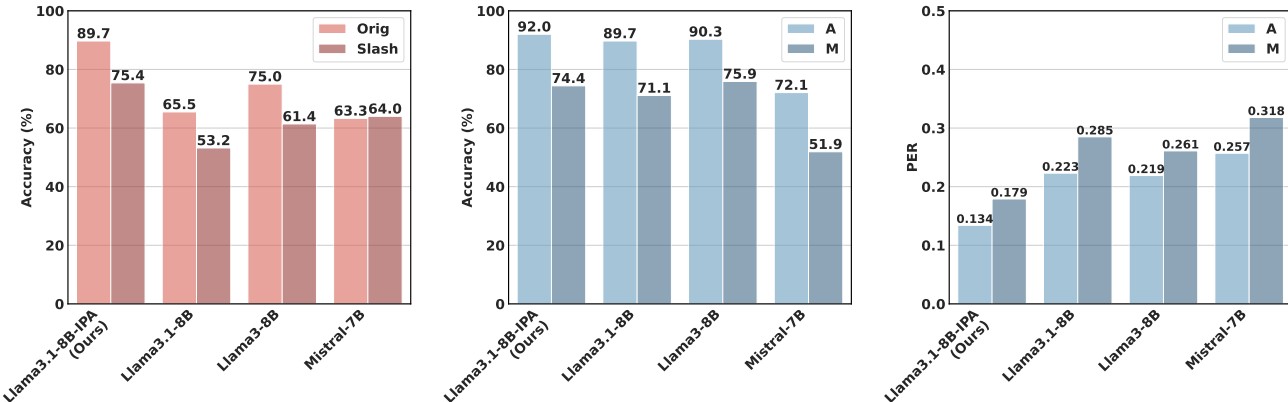

*Figure 2.* Performance comparison of three language models (Llama3.1-8B, Llama3-8B, and Mistral-7B) and the LoRA-fine-tuned Llama3.1-8B-IPA model on three phonology-related tasks. The **left** figure corresponds to the Rhyming Awareness task (higher accuracy is better), the **middle** figure corresponds to the Syllable Counting task (higher accuracy is better), and the **right** figure corresponds to the Grapheme-to-Phoneme (G2P) task (lower phoneme error rate (PER) is better). The fine-tuned Llama3.1-8B-IPA model achieves the highest accuracy in Rhyming Awareness and Syllable Counting while also attaining the lowest PER in the G2P task, demonstrating the effectiveness of IPA fine-tuning for phonological reasoning.

For a given general QA conversation, we augment it by randomly selecting 0–2 words in the question and wrapping the word using IPA token. And in the beginning of the answer, we add one sentence giving the IPA of the chosen words. If no word is chosen, we do not add anything to the original data. For the domain-specific data, we also conversationally create the data, where questions are a series of possible prompts, and answers are an inference process using the IPA (see Appendix B for an example).

| Data Type | Number of Examples |
| --- | --- |
| Conversation | 3000 |
| Rhyming Awareness | 200 |
| Syllable Counting | 500 |
| G2P | 500 |

*Table 3.* Number of fine-tuning examples from each source.

### 4.2. Experiment Setup

**Dataset.** Our dataset consists of two sources: (1) high-quality general instruction-tuning conversation data sampled from OpenHermes2.5 (Teknium, 2023), and (2) phonology-related tasks, where we select words **outside** the Google-10000-English dataset as training examples. We summarize the data statistics in Table 3, and present the detailed data construction template in Appendix B.

**Models.** We evaluate three tasks using the dataset described in Section 3, conducting zero-shot inference, with prompt templates provided in the appendix. We assess three instruction-tuned LLMs: Llama3.1-8B, Llama3-8B, and Mistral-7B-v3. Furthermore, we fine-tune the Llama3.1-8B

model using LoRA (Hu et al., 2021) on our constructed instruction-tuning dataset (see Table 3).

**Evaluation.** For the rhyming awareness task, we extract the model's true/false predictions and compute the accuracy. For the syllable counting task, we evaluate the model's predicted syllable count against the ground truth, and also report the accuracy. For the G2P task, we compute the Phoneme Error Rate (PER), defined as the Levenshtein distance between the predicted and reference phoneme sequences (both in ARPAbet), normalized by the number of phonemes in the reference pronunciation. A lower PER indicates better transcription quality. The results for all three tasks are presented in Figure 2. We present the prompt used for evaluation in Appendix C.

To ensure there is no catastrophic forgetting in our fine-tuned model, we also evaluate it on two widely used benchmarks: GSM8K (Cobbe et al., 2021) and MMLU (Hendrycks et al., 2021). We employ a chat-style zero-shot evaluation to simulate real-world user interactions. For MMLU, we randomly sample three subjects from each major category (STEM, social sciences, humanities, and other), totaling 12 subjects.

### 4.3. Results

As presented in Figure 2, for the rhyming awareness task, we find that simply inserting slashes into words does not necessarily improve performance; instead, it decreases inference accuracy. We hypothesize that this occurs because the slashes disrupt the tokenization structure of the input, leading to incorrect predictions. This observation suggests that tokenization is not the sole factor influencing model decisions in rhyming tasks—other factors, such as the statis-

| Model | GSM8K | MMLU |
|---|---|---|
| Llama3.1-8B-Instruct | 70.4 | 65.3 |
| Llama3.1-8B-IPA (Ours) | 67.9 | 64.4 |

*Table 4.* Performance comparison between the Llama3.1-8B-Instruct model and the fine-tuned Llama3.1-8B-IPA model on non-phonology tasks. The reported metrics are accuracy for both GSM8K (math reasoning) and MMLU (general knowledge and understanding). Fine-tuning with IPA results in only a minor performance drop in general tasks, indicating effective knowledge retention.

tical distribution of words in the training corpus, may also play a role. However, by incorporating IPA representations, model performance improves significantly, achieving nearly 90% accuracy on our rhyming awareness dataset.

For the two prosodic structure tasks, LMs perform considerably better on words with low STAD scores, indicating that, beyond learned representations, tokenization structure also affects inference quality. Our fine-tuned IPA model significantly enhances performance on the G2P task. However, the improvement in syllable counting for low-STAD words remains relatively minor, as the baseline Llama3.1-8B-Instruct model already demonstrates strong proficiency in this task. On the other hand, the improvement for high-STAD words is more pronounced, suggesting that fine-tuning with IPA reduces the bias introduced by tokenization. Nevertheless, despite fine-tuning the model for phonology-related tasks using IPA, some degree of bias introduced by tokenization remains.

Due to the way we curated the dataset, our model maintains strong performance in phonology-related tasks while largely preserving its capabilities in other domains (Table 4). The evaluation of GSM8K and MMLU shows that the fine-tuned model, Llama3.1-8B-IPA, retains most of its general reasoning and knowledge abilities. Specifically, compared to the original Llama3.1-8B-Instruct model, the accuracy drop is minimal—only 2.5 percentage points on GSM8K and 0.9 percentage points on MMLU. This demonstrates that our fine-tuning approach effectively enhances phonology-related reasoning without significantly compromising performance on broader language understanding and reasoning tasks.

## 5. Cognates Based Analysis

Having identified the potential biases that tokenization introduces to language models (LMs) in phonology-related tasks, we now investigate the underlying causes and the types of words most susceptible to such tokenization discrepancies. Most modern tokenization algorithms, such as BPE and SentencePiece, optimize subword segmentation to maximize

corpus frequency or model likelihood. Consequently, words whose tokenization misaligns with their natural syllabification often exhibit significant orthographic variability in the training corpus, arising from historical processes such as lexical borrowing and etymological divergence.

A key factor contributing to such variation is the presence of **cognates** and **loanwords** across languages. To systematically examine this phenomenon, we use **CogNet** (Batsuren et al., 2019), a comprehensive database of cognate words and loanwords, to identify potential cognates associated with a given lexical item.

To empirically test whether the presence of cognates correlates with deviations in syllabification-aligned tokenization, we analyze the average number of cognate words for aligned words (A) and misaligned words (M) across six different tokenizers, as discussed in Section 3.2. The results, reported in Figure 3, indicate that words in group M systematically exhibit a higher number of cognate variants than those in group A across all evaluated tokenizers. This finding suggests that words with extensive cross-linguistic cognacy are more likely to undergo non-standard tokenization.

From a linguistic perspective, this correlation can be explained by the fact that words with a greater number of cognates tend to be semantically and morphologically richer.

Such words often undergo multiple layers of phonological and orthographic adaptation across languages, leading to greater variability in their written forms. This variability increases the likelihood that tokenization algorithms will segment them in ways that diverge from their natural phonological structure. Additionally, because tokenization algorithms prioritize frequency-based segmentation, highly polysemous or widely borrowed words may be tokenized in ways that reflect corpus-level distribution rather than phonological intuition.

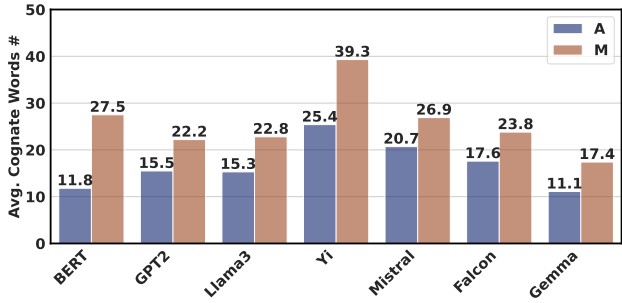

*Figure 3.* Average number of cognates of token-syllable aligned words (A) and token-syllable misaligned words (M) for different tokenizers.

# 6. Conclusion and Discussion

We have evaluated three phonology-related tasks centered on local phonological coherence and prosodic structure, demonstrating that the tokenization techniques used by LMs can introduce biases in word representation, thereby limiting their performance on these tasks. To address this challenge, we have proposed an efficient approach that leverages a small amount of data and computational resources to enhance LM performance. Additionally, we identify a correlation between tokenization bias and the linguistic variability of words, though the causal relationship remains an open question.

Insights from this work may also benefit the development of joint speech and text language models (e.g., Chou et al., 2023, *inter alia*), by enabling better text tokenization that preserves nuanced phonological information.

Finally, it is worth noting that our experiments have been focused on English, a representative alphabetic language. Findings in this work need significant work to be possibly adaptable to logographic languages. We leave the exploration of a broader range of modal architectures and additional languages for future work.

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

# A. Controlled-Experiment of the Probing

To verify that our linear probes do not artificially inflate performance, we repeat every probing experiment with *randomly generated targets*. For rhyming awareness, we assign a random binary label to each word-pair; for grapheme-to-phoneme (G2P) prediction we draw a random integer in the range $[0, 39]$ for every phoneme slot; and for syllable counting, we sample a random integer between 0 and 8. We train the same logistic- and ridge-regression probes as in the main study on GPT-2 and Llama-3.1-7B using these synthetic labels.

As summarised in Figure 4, the control probes behave exactly as expected: accuracy hovers around the chance rate of 0.5 for the binary classification task, and all $R^2$ values for the two regression tasks are zero or negative across layers. This confirms that the probes themselves lack the capacity to memorize arbitrary labellings and that the positive results reported in the main paper genuinely stem from information encoded in the models' hidden representations rather than from overfitting artefacts.

# B. Phonology-related Task Training Template

---

**Algorithm 1** Conversation Data Creation with IPA Annotations

---

**Require:** Dataset $\mathcal{D}$ with question-answer pairs $(q, a)$, IPA sentence template $L$. Function `fill`$(T, w)$ to fill a template $T$ using word $w$. Function `get_IPA` to get IPA.
**Ensure:** Modified dataset $\mathcal{D}'$ with IPA-annotated questions and answers
1: $\mathcal{D}' \leftarrow \emptyset$
2: **for** each $(q, a) \in \mathcal{D}$ **do**
3:     Split $q$ into words: $W \leftarrow \text{split}(q)$
4:     Sample $k \sim \text{Uniform}(\{0, 1, 2\})$
5:     Uniformly sample $S \subset W$ with $|S| = k$
6:     **for** each selected word $w_i \in S$ **do**
7:         Replace $w_i$ in $q$ with $\langle\text{IPA}\rangle$ $w_i$ $\langle/\text{IPA}\rangle$.
8:         Obtain IPA transcription of $w_i$: $I = \text{get\_IPA}(w_i)$
9:     **end for**
10:     filling in words from $S$ into $L$, $l = \text{fill}(L, S)$
11:     Prepend sentence $l$ indicating IPA representation to $a$: $a' \leftarrow a + l$
12:     Add modified pair $(q', a')$ to $\mathcal{D}'$
13: **end forreturn** $\mathcal{D}'$

---

Here, we demonstrate the detailed training template we used for constructing the training dataset for each problem. We demonstrate how we construct 4 categories of QA pairs as our fine-tuning dataset.

**Conversation.** We used OpenHermes2.5 (Teknium, 2023) as the source of the conversation dataset, it involves all kinds of conversational datasets consisting of question and answer. In the question, we randomly select 0 - 2 words and wrap the words with an IPA token to indicate that we want the IPA of the word, and in the answer, we add one sentence

---

**Algorithm 2** Dataset Creation for Rhyming Awareness Task

---

**Require:** Word pair list with IPA transcriptions, possible templates $P_1, \ldots, P_5$. Positve answer template $A_P$, negative answer template $A_N$. Function `fill`$(T, w)$ to fill a template $T$ using word $w$.
**Ensure:** Dataset with question-answer pairs
1: **for** each $(word_1, word_2)$ pair **do**
2:     Sample $i \sim \text{Uniform}(\{0, 1, 2, 3, 4, 5\})$
3:     $P \leftarrow \text{fill}(P, (word_1, word_2))$.
4:     Extract IPA endings of $word_1$ and $word_2$
5:     **if** IPA endings match **then**
6:         response $\leftarrow \text{fill}(A_P, (word_1, word_2))$
7:     **else**
8:         response $\leftarrow \text{fill}(A_N, (word_1, word_2))$
9:     **end if**
10:     Store $(P_i, \text{response})$ in dataset
11: **end for**

---

**Algorithm 3** Dataset Creation for Grapheme-to-Phoneme (G2P) Task

---

**Require:** Word list with IPA transcriptions, possible templates $P_1, \ldots, P_5$. Answer template $A$. ARPAbet-to-phoneme dictionary $M$. Function `fill`$(T, w)$ to fill a template $T$ using word $w$.
**Ensure:** Dataset with question-answer pairs
1: **for** each word $w$ in dataset **do**
2:     Sample $i \sim \text{Uniform}(\{0, 1, 2, 3, 4, 5\})$
3:     $P \leftarrow \text{fill}(P_i, w)$
4:     Obtain IPA transcription of $w$: $I = \text{get\_IPA}(w)$
5:     IPA phonemes to ARPAbet: $A = [M[p] \text{ for } p \in I]$
6:     response $\leftarrow \text{fill}(A, (w, I, A))$
7:     Store $(P, \text{response})$ in dataset
8: **end for**

---

**Algorithm 4** Dataset Creation for Syllable Counting Task

---

**Require:** Word list with IPA transcriptions, possible templates $P_1, \ldots, P_5$. Answer template $A$. Function `fill`$(T, w)$ to fill a template $T$ using word $w$.
**Ensure:** Dataset with question-answer pairs
1: **for** each word $w$ in dataset **do**
2:     Sample $i \sim \text{Uniform}(\{0, 1, 2, 3, 4, 5\})$
3:     $P \leftarrow \text{fill}(P_i, w)$
4:     Obtain IPA transcription of $w$: $I = \text{get\_IPA}(w)$
5:     Identify vowels and diphthongs in $I$
6:     Compute syllable count: $S = \text{count\_syllables}(I)$
7:     Format response using $S$: response $\leftarrow \text{fill}(A, (w, S))$
8:     Store $(P, \text{response})$ in dataset
9: **end for**

---

indicating the IPA of the words. We present the process of data creation in Algorithm 1 and show an example in Figure 5a.

**Rhyming Awareness.** In the rhyming awareness, we prepare 5 possible question templates $P_1, P_2, \ldots P_5$ to mimic the possible users' questions. In the answer, we first give the IPA of the word as in Conversation. Then, from the IPA, we extract the same part of the IPA if two words are in rhyme, or state two words are not in rhyme if the IPA

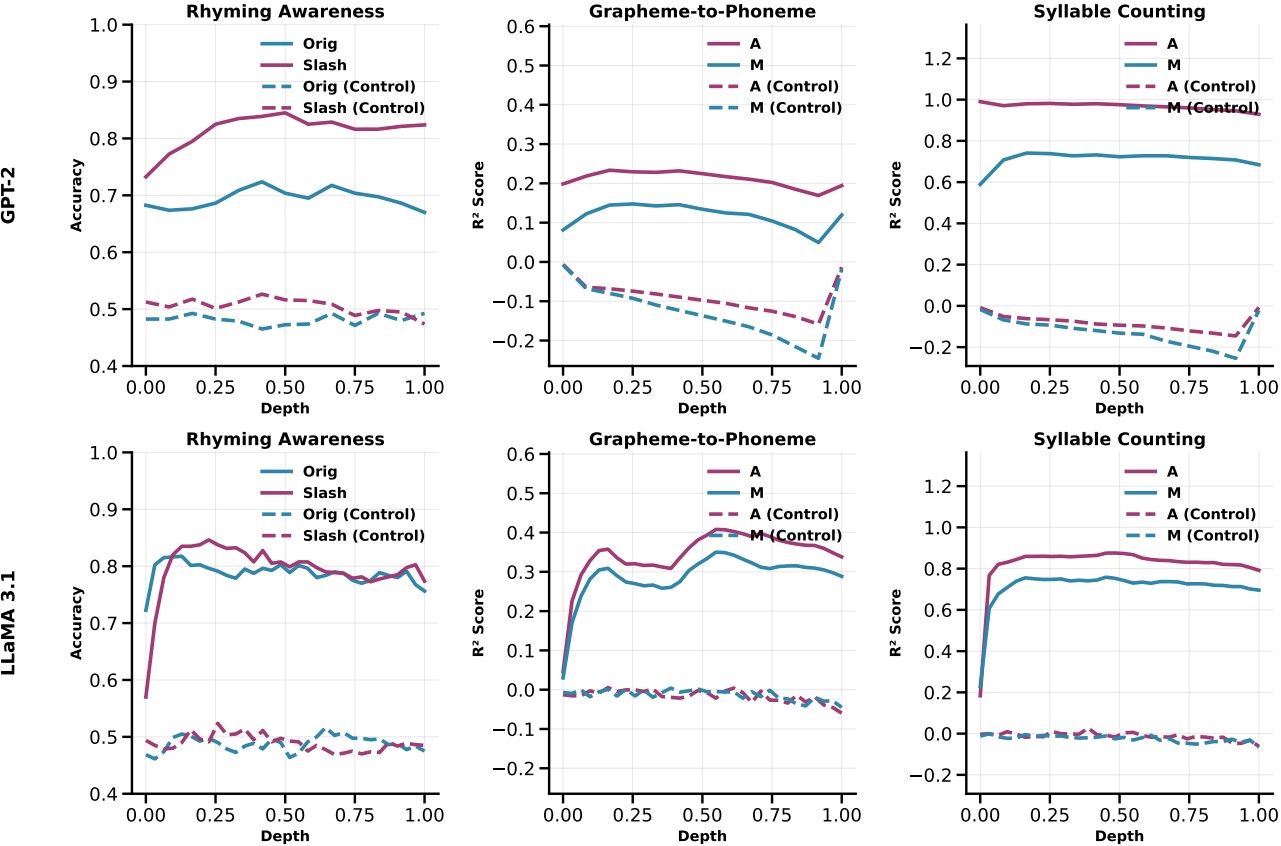

*Figure 4.* **Control-label sanity check.** Probing performance with *random* targets for GPT-2 (upper block) and Llama-3.1-7B (lower block). Left-to-right: (i) Rhyming awareness—accuracy; (ii) G2P—$R^2$; (iii) syllable counting—$R^2$. Solid lines reproduce the original probes, dashed lines the corresponding control probes. All curves collapse to chance (accuracy $\approx 0.5$) or sub-chance ($R^2 \leq 0$), demonstrating that the linear probes do not overfit when the target carries no linguistic signal.

does not have the same ending. We present the process of rhyming awareness data creation in Algorithm 0 and we show the question templates and example answer in Figure 5b.

**G2P.** In the G2P task, we also prepare 5 possible question templates. In the answer, we break the IPA transcript phoneme by phoneme and use the map from phoneme to ARPAbet to convert the IPA to ARPAbet. We present the process of G2P data creation in Algorithm 0 and the question templates and an example answer in Figure 5c.

**Syllable Counting.** In the syllable counting, there are also 5 possible questions as before, in the solution, we first identify the vowels and diphthongs in the IPA transcript and add up the number of vowels and diphthongs, which will give the total number of syllables. We present the process of syllable counting data creation in Algorithm 0 and the question templates and an example answer in Figure 5d.

## C. Evaluate Prompt For Phonology Inference

To assess phonological competence in a controlled, model-agnostic manner, we rely on three single-turn, zero-shot prompts, one for each task shown in Figure 6. Each template first frames the task in plain English—defining either the ARPAbet symbol inventory, the notion of rhyme, or the meaning of a syllable—before requesting an answer in an unambiguous stub such as *ARPAbet: . . . , Answer: True/False*, or *Answer: <number>*. By stipulating the permissible symbol set and response syntax up front, the prompts eliminate formatting variance and allow us to grade outputs with exact matching rather than heuristics. The resulting protocol isolates the models' ability to map orthography to phonological structure, guaranteeing that any performance differences reported in the main paper stem from representational content rather than prompt interpretation or output-parsing artefacts.

Q:**<IPA>Every</IPA>** *day, a tree drops 7 leaves. How many* **<IPA>leaves</IPA>** *would it drop in a month of February in a non-leap year? Include your logic.*

A: **word Every has IPA /ˈɛv.(ə.)ɹi/ and leaves has IPA /liːvz/.**
Here's the logic behind this:\n\n1. We know that February has 28 days in a non-leap year.
2. If the tree drops 7 leaves every day, then over the course of February, it would drop: Leaves dropped in February = Leaves per day * Days in February = 7 leaves * 28 days = 196 leaves.
So, the tree would drop 196 leaves in February in a non-leap year

(a) An example of how we construct QA data from OpenHermes2.5.

**P1**: *Consider words* <IPA>{word1}</IPA> *and* <IPA>{word2}</IPA>. *Do they rhyme?*
**P2**: *Are the following words in rhyme?* <IPA>{word1}</IPA>, <IPA>{word2}</IPA>?
**P3**: *Does word* <IPA>{word1}</IPA> *rhyme with word* <IPA>{word2}</IPA>?
**P4**: *Is word* <IPA>{word1}</IPA> *and* <IPA>{word2}</IPA> *in rhyme?*
**P5**: *Rhyming words are words that have the same ending sound. Is word* <IPA>{word1}</IPA> *in rhyme with word* <IPA>{word2}</IPA>?

**Example Answer 1**: cat has IPA /kæt/ and hat has IPA /hæt/.
From the IPA transcriptions, cat and hat have the same ending sound /æt/, therefore they are in rhyme.
Answer: Yes
**Example Answer 2** : rain has IPA /ɹeɪn/ and bloom has IPA /bluːm/.
From the IPA transcriptions, rain and bloom have different ending sounds, therefore they are not in rhyme.
Answer: No

(b) All possible questions and example answers fo rhyming awareness task.

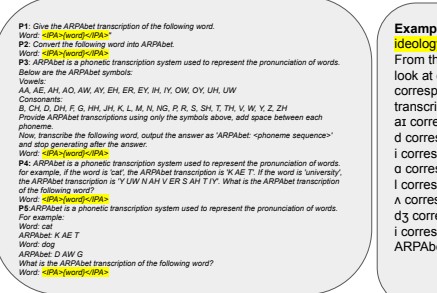

(c) All possible questions and an example answer for G2P task.

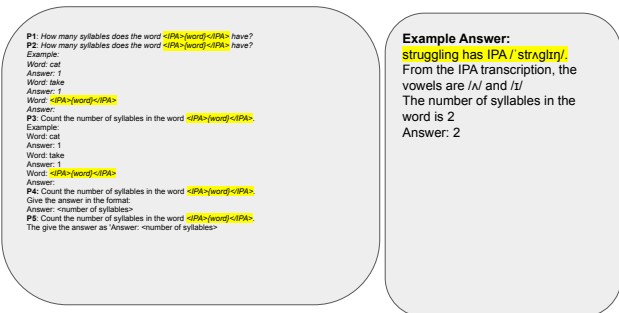

(d) All possible questions and an example answer for syllable counting task.

*Figure 5.* Examples of our question templates and some example answers. The yellow part is the common part of the fine-tuning dataset, which helps the model to identify which word to consider IPA and give the IPA explicitly.

**G2P Prompt**

ARPAbet is a phonetic transcription system used to represent the pronunciation of words Below are the ARPAbet symbols

Vowels:
AA, AE, AH, AO, AW, AY, EH, ER, EY, IH, IY, OW, OY, UH, UW

Consonats:
B, CH, D, DH, F, G, HH, JH, K, L, M, N, NG, P, R, S, SH, T, TH, V, W, Y, Z, ZH

Provide ARPAbet transcription using only the symbols above, add space between each phoneme. Transcribe the following word, output the answer as 'ARPAbet: <phoneme sequence>'.
Word: {word}

(a) The prompt template we used to evaluate the G2P task.

**Rhyming Awareness Prompt**

Rhyming words are words that have the same ending sound. Determine if the following two words are in Rhyme.
{word1}, {word2}
Give the answer as "Answer: True" if they rhyme and "Answer: False" if they do not.

(b) The prompt template we used to evaluate the rhyming awareness task.

**Syllable Counting Prompt**

Count the number of syllables in the word: '{word}'
Give the answer in the format "Answer: <number of syllables>"

(c) The prompt template we used to evaluate the syllable counting task.

*Figure 6.* Zero-shot prompt templates for (a) G2P transcription, (b) rhyme judgement, and (c) syllable counting, each with a fixed answer stub for scoring.

## D. Probing Details

If we have $n$ words/pairs of words as input, after prompting them to LMs, for each layer $l$, we will get a matrix of $\boldsymbol{H}_l \in \mathbb{R}^{n \times d}$, where $d$ is the dimension of the hidden states. Then, if we have the ground truth $\boldsymbol{y}$, we can train models using $\boldsymbol{H}_l$ and $\boldsymbol{y}$, and we will discuss the details of our probing for each task. For the model we trained, we used scikit-learn (Pedregosa et al., 2011) implementation. For

each experiment, we ran 10 times with seeds 0 - 9, and did an 80 - 20 train-test split, reporting the metrics on the test set. We only selected a linear model for evaluation, since the goal of our work is not for achieving high performance on the downstream task but to illustrate the bias introduced by the tokenizers. Also, Hewitt & Liang (2019) illustrated that using a complex model like Neural Network may cause the probing result unreliable since the model will learn the feature, and our evaluated tasks are not a very hard task, thus, the linear model is enough to reveal the representation quality of different words.

**Rhyming Awareness**. For the rhyming awareness task, the input is a pair of words, and ground truth $y \in \mathbb{R}^n$ is a binary label indicating if the words pair is rhyming. Then we used logistic regression `LogisticRegression`, and we set the max iterations to 1000, and Inverse of regularization strength $C = 10$, other hyperparameters are set as default. We trained two logistic regression classifiers on both original words and words with slash inserted.

**G2P.** For the G2P task, the label is the categorical encoding of the ARPAbet symbols (0 - 39), we either truncated or padded the label with 0. Therefore, we have $Y \in \mathbb{R}^{n \times 8}$. We used the Cross-validation Ridge regression `RidgeCV` to regress the label and set the alphas to be chosen from $\{10, 100, 500, 1000, 2000\}$, other hyperparameters are set as default. We train two ridge regressors on both syllable-token aligned and misaligned groups.

**Syllable Counting.** For the syllable counting task, the label is the number of syllables in the word. Therefore, we have $y \in \mathbb{R}^n$. We also used Cross-validation Ridge regression `RidgeCV` to regress the label and set alphas to be chosen from $\{10, 100, 500, 1000, 2000\}$, other hyperparameters are set as default. We train two ridge regressors on both syllable-token aligned and misaligned groups.

## E. Fine-tuning & Evaluation Details

For the evaluation, we used the chat template of the corresponding model to form the QA. And we used vllm (Kwon et al., 2023) and set the decoding strategy to greedy.

For the fine-tuning, we leveraged the Hugging Face `transformers` library along side Parameter-Efficient Fine-Tuning (PEFT) to integrate LoRA (Hu et al., 2021). We specifically targeted the query (q_proj) and value (v_proj) projection layers for adaptation. We set the LoRA Rank ($r$) to 8, LoRA scaling factor ($\alpha$) to 16, LoRA Dropout to 0.1.

For multi-GPU training, we employed Hugging Face Accelerate, which facilitated seamless distributed training across the two GPUs using Pytorch Distributed Data Parallel (DDP). The model and dataset were automatically partitioned and synchronized, ensuring efficient computation.

| Model Size | Delimiter | Depth (Accuracy ↑) | | | | | |
|---|---|---|---|---|---|---|---|
| | | 0% | 20% | 40% | 60% | 80% | 100% |
| *Sub-word Tokenization* | | | | | | | |
| **BERT-110M** | | | | | | | |
| 110M | None | 56.0 | 67.6 | 68.3 | 70.9 | 71.0 | 70.5 |
| | Slash | 68.6 | 74.5 | 73.4 | 77.5 | 79.5 | 78.1 |
| | Comma | 68.6 | 75.7 | 71.8 | 77.8 | 80.3 | 79.2 |
| | Dot | 68.6 | 74.7 | 73.6 | 80.8 | 79.8 | 78.4 |
| **GPT-2-1.2B** | | | | | | | |
| 1.2B | None | 63.4 | 64.7 | 66.1 | 66.2 | 66.0 | 61.6 |
| | Slash | 71.7 | 76.9 | 77.2 | 79.1 | 78.5 | 77.5 |
| | Comma | 71.7 | 77.6 | 77.1 | 78.8 | 77.9 | 77.3 |
| | Dot | 72.3 | 77.3 | 77.6 | 79.5 | 78.4 | 76.9 |
| **Llama-3.1-8B-Instruct** | | | | | | | |
| 8B | None | 72.5 | 79.8 | 79.0 | 77.9 | 77.3 | 74.9 |
| | Slash | 56.3 | 85.1 | 84.0 | 80.0 | 78.9 | 79.5 |
| | Comma | 56.8 | 86.7 | 83.7 | 79.3 | 78.5 | 77.4 |
| | Dot | 56.7 | 85.2 | 82.0 | 79.8 | 77.8 | 76.3 |
| **Mistral-7B-Instruct-v3** | | | | | | | |
| 7B | None | 64.5 | 80.6 | 80.8 | 78.8 | 77.4 | 74.7 |
| | Slash | 55.8 | 81.1 | 82.7 | 79.5 | 79.0 | 77.6 |
| | Comma | 55.8 | 81.7 | 85.4 | 82.1 | 80.3 | 79.5 |
| | Dot | 55.8 | 80.5 | 81.7 | 79.1 | 77.9 | 77.2 |

*Table 5.* Ablation study of delimiter formats ("None", "Slash", "Comma", "Dot") across different depths of the hidden states.

## F. Rhyming probing using Different Delimiters

In the rhyming awareness probing experiment, we initially used the slash ("\") as a delimiter to split word pairs, enabling more structured and representative hidden states. To assess the robustness of this delimiter choice—and to test whether performance gains stem from improved tokenization granularity rather than the specific symbol—we conducted an ablation study using alternative delimiters: the comma (",") and the dot (".").

We evaluated probing performance across four language models—BERT, GPT-2, LLaMA3.1-8B, and Mistral-7B—and report results in Table 5. Across all models, the probers trained with any delimiter (slash, comma, or dot) yield comparable performance throughout the depth of the hidden layers. Importantly, all delimiter-based variants consistently outperform the baseline where no delimiter is used (None), confirming that the performance gains are primarily due to the introduction of fine-grained structure in the input rather than the specific choice of delimiter.

