# OpenReview forum: "How Tokenization Limits Phonological Knowledge Representation in Language Models and How to Improve Them"
_ICML.cc/2025/Workshop/TokShop — TokShop_

### Official Review · Reviewer_4xEX · 2025-06-07
**Very interesting paper and results but very incoherently written**

**Rating:** 6
**Confidence:** 4

**Review:**

Summary/Strengths
This paper makes 3 contributions:

1. It probes hidden representations of LLMs with different tokenization schemes to understand how tokenization affects the model’s encoding of phonological information, information relating to grapheme to phoneme, and information relating to syllabic awareness.
2. It introduces a metric called STAD score to measure the discrepancy between tokenization and syllabification for different LLM tokenizers.
3. It introduces a fine-tuning dataset using IPA annotations, that improves the 0-shot performance of current models on tasks relating to rhyme identification, phoneme generation etc. without hurting model performance on regular text benchmarks.

First & Second:
The paper probes the final hidden representations of LLMs for phonological information using three tasks: 1) rhyming awareness, 2) Grapheme to Phoneme, and 3) Syllable Counting.

For 1) they develop a new dataset of 200 +ive and 200 -ive examples (320 training and 80 test), taking care that endings are different, so that the model cannot cheat. They evaluate a comprehensive set of token and byte level LLMs upto 8B in size. They also test a variant where they insert '/' between the characters to force the model to tokenize the characters separately i.e. boy --> b/o/y and find that the model with / vastly outperforms the model without, with the gap being closed for instruct models. They conclude that finer-grained tokenization aids in uncovering phonological information from the model. They also make observations that earlier layers are better predictors of phonological information, and that byte level models are better on this task as well.

The paper introduces a new metric called the STAD score, which measures the discrepancy between its tokenization and syllabification. This metric is the hamming distance between the binary representation of the tokenization split and the syllabification split of a word. For the G2P task, the paper takes 1000 words that are syllabically misaligned, and 1000 words that are perfectly aligned, for each LM. Then it passes the word through the LM, records the hidden states, and tried to use ridge regression to predict the phoneme, which is encoded as a 8-dimensional vector in ARPAbet format.  Similarly, they also do a syllable counting regression task.

Third:
The paper introduces some kind of fine-tuning dataset with IPA annotations, which improves the performance on the 3 tasks mentioned above using 0-shot prompt-based evaluation, when fine-tuned on the dataset. This also does not degrade performance on existing text benchmarks.

Overall: The paper has very interesting findings of how tokenization affects the models encoding of phonological knowledge, and how it can be improved by fine-tuning with IPA annotations.

Weaknesses/Questions

Overall: The paper is written in a very confusing way. First results of the rhyming task are presented. Then a different task is described. Then the first results are discussed, with no discussion about the second results. All important examples are move to the appendix. The fine-tuning dataset is described without any examples. The paper lacks a coherent flow and has been very cumbersome to review. The content of the paper is very interesting but a heavy cognitive overload to understand. I recommend the authors ask a non-author to read the paper and make sure it flows coherently. Since its a workshop, I recommend a weak accept, but would like answers to the following questions.

Here are some questions:

1. One question/concern about the setup of adding slashes to separate tokens for rhyming, is the use of different amounts of inference compute. The model with / spends double the compute for each hidden representation, and it may be that which is causing better performance. For open models like Llama3, it's possible to force single character tokenization without inserting /. Why did the authors not use this approach, which doesn't cause a bias in terms of compute usage?

2. The STAD score does not take into account the degree of discrepancy. For example, the word hamming, with syllable split = ['ham', 'ming'] =001000 with achieve the same score for ['h', 'amming'] = 100000, ['ha', 'ming'] = 010000, ['hammi', 'ng']  = 000010. Are these splits considered equivalent? If so, why?

3. I could not find a discussion of the results in Table 2 i.e. for the G2P and Syllable tasks - just a very large table. Could you authors include a discussion of these results?

4. The paper suddenly refers to a QA dataset on line 287 column 2, without introducing where QA comes from in this context suddenly.

5. What is the meaning of “wrapping the word using the IPA token”. An example of this would be great.

6. Why was MMLU not evaluated fully? And why not 5-shot which is standard?

7. In Figure 2, which is Mistral not finetuned with IPA? What purpose does it serve in the table? Are IPA representations used in the evaluation prompt for the IPA setting?

---

### Decision · Program_Chairs · 2025-06-10

Accept